# Early Plasma Magnesium in Near-Term and Term Infants with Neonatal Encephalopathy in the Context of Perinatal Asphyxia

**DOI:** 10.3390/children9081233

**Published:** 2022-08-15

**Authors:** Juliette Suhard, Cathie Faussat, Baptiste Morel, Emeline Laurent, Geraldine Favrais

**Affiliations:** 1Neonatology Department, Regional University Hospital Centre Tours, 37000 Tours, France; 2CoMeth Team, Public Health Department, Regional University Hospital Centre Tours, 37000 Tours, France; 3Pediatric Radiology Department, Regional University Hospital Centre Tours, 37000 Tours, France; 4UMR 1253, iBrain, Université de Tours, INSERM, 37044 Tours, France

**Keywords:** magnesium, hypoxia-ischaemia, brain, brain injuries, infant, newborn

## Abstract

Magnesium ions are implicated in brain functioning. The disruption of brain metabolism subsequent to a perinatal hypoxic-ischaemic insult may be reflected by plasma magnesium. Infants at 36 weeks after birth or later with neonatal encephalopathy and who were admitted to our neonatal unit from 2011 to 2019 were retrospectively included. The kinetics of plasma magnesium were investigated for the first 72 h of life and correlated to the Barkovich MRI score. Among the 125 infants who met the inclusion criteria, 45 patients (36%) had moderate to severe brain lesions on neonatal MRI. Plasma magnesium values were not strongly associated with the severity of clinical encephalopathy, initial EEG background and brain lesions. Intriguingly, higher plasma magnesium values during the 0–6 h period were linked to the presence of brain injuries that predominated within the white matter (*p* < 0.001) and to the requirement of cardiac resuscitation in the delivery room (*p* = 0.001). The occurrence of seizures was associated with a lower mean magnesium value around the 24th hour of life (*p* = 0.005). This study supports that neonatal encephalopathy is a complex and multifactorial condition. Plasma magnesium could help to better identify the subtypes of neonatal encephalopathy. Further studies are needed to confirm these results in this prospect.

## 1. Introduction

Neonatal encephalopathy (NE) in newborn infants can result from perinatal asphyxia. The incidence of NE in this context is approximately 1–3.5 per 1000 live births in developed countries [1].

Hypoxic-ischaemic insult during the perinatal period in near-term and term infants induces characteristic brain damage. Although therapeutic hypothermia (TH) is associated with the reduction of hypoxic-ischaemic brain lesions, these injury patterns were not modified by TH [2]. The deep grey matter, i.e., basal ganglia and thalami as well as the posterior limb of the internal capsule, and the watershed areas of the white matter are particularly vulnerable to a perinatal hypoxic-ischaemic event [3]. Interestingly, the severity of injury concerning these specific brain areas has been associated with the neurological outcome of infants with NE [2]. Therefore, MRI scorings such as the Barkovich score have been developed to support the assessment of neurological prognosis after the completion of TH [4].

Despite the implementation of TH as a standard of care, concerns remain about the prognosis of infants with NE. The mortality rate is approximately 30%, and cerebral palsy is diagnosed in 17–21% of infants who survive NE [5,6]. The prospects to improve NE outcomes have focused on early and bedside monitoring and potential new neuroprotective agents that may be added to TH. This monitoring may include perinatal biomarkers, electrophysiology, cerebral perfusion and oxygenation assessment. This approach aims to make care more appropriate to the condition of each infant. This could be useful for better identifying infants who could fully benefit from TH, for better understanding the development of hypoxic-ischaemic brain lesions and for adjusting neuroprotective strategies [7,8].

Magnesium ions are implicated in brain functioning through membrane stabilization, mitochondrial function, and anti-inflammatory and anti-excitotoxic actions [9,10,11,12]. Experimental studies have shown that brain free magnesium and blood magnesium decrease after severe traumatic brain injury [13,14]. In humans, Ilves et al. reported alterations of plasma magnesium in non-cooled infants with severe NE [15,16]. These results suggested that plasma magnesium is closely linked to the severity of brain dysfunction following a hypoxic-ischaemic insult. Brain cellular responses to a hypoxic-ischaemic insult have been characterized in experimental models. Immediate cell death is observed due to the drop in cerebral perfusion and oxygenation during the hypoxic-ischaemic challenge and corresponds to the primary phase. After resuscitation and the restoration of cerebral perfusion, cerebral hypoperfusion associated with a partial improvement in brain functioning is observed, corresponding to the latent phase. Subsequently, a secondary energy failure occurs. This secondary phase is characterized by cerebral hyperperfusion, mitochondrial dysfunction, delayed cell death and the possible occurrence of seizures. This phase begins 6 to 12 h after insult and can persist for several days. TH prevented this deleterious cascade in experimental models, supporting its neuroprotective effect and justifying its optimal implementation before the sixth hour of life [17]. Finally, recovery and repair mechanisms appear and correspond to the tertiary phase [8].

In this study, we speculated that the disruption of brain metabolism subsequent to a perinatal hypoxic-ischemic insult could be reflected by the alteration of plasma magnesium kinetics during the early phase of NE and that the plasma magnesium profile may be specific to the severity of the brain lesions on MRI and the injured brain areas. Therefore, our primary aim was to depict the kinetics of plasma magnesium over the first 72 h of life according to the severity of brain lesions and the predominant injured brain area in a recent cohort of infants with NE following perinatal asphyxia. We also explored the perinatal factors related to perinatal asphyxia and NE that could alter plasma magnesium values.

## 2. Materials and Methods

### 2.1. Participants

Study eligibility was restricted to infants who were hospitalized in the neonatal unit of the University Hospital of Tours, France, from 2011 to 2019 and diagnosed with NE with suspected hypoxic-ischaemic origin. The term NE is used throughout this article in preference to hypoxic-ischaemic encephalopathy due to the lack of evidence of isolated, peripartum and acute hypoxic-ischaemic challenge in all cases of our cohort and following the recommendation of key opinion leaders on this topic [18]. However, the authors assume that the NE cases selected for this study were NE with a presumed hypoxic-ischaemic origin. All medical files were reviewed (JS, GF). The noninclusion criteria were gestational age less than 36 weeks, absence of objective elements supporting perinatal asphyxia, sudden and unexpected postnatal collapse, absence of brain magnetic resonance imaging (MRI), and no plasma magnesium measurement for the first 72 h of life. The exclusion criterion was the opposition by parents to the utilization of their child’s data.

At least one of the following three criteria was required for the validation of perinatal asphyxia. The first criterion was the presence of blood metabolic acidosis (i.e., pH < 7.1 or base-deficit > −11 mmol/L) in the umbilical cord artery or during the first hour of life. The second criterion was the presence of signs during labour that revealed a threat to foetal well-being (i.e., foetal heart rate abnormalities, meconium amniotic flood, clinical chorioamnionitis or decrease in foetal active movements), emergent C-section or instrumental vaginal delivery. The third criterion was an identified sentinel event (e.g., placental abruption, cord prolapse, uterine rupture). Acute perinatal asphyxia was defined by the presence of a sentinel event that clearly explained the origin of asphyxia without any complications during pregnancy or the hours preceding delivery. Recent foetal heart rate abnormalities and emergent C-section could be associated with this context.

NE was classified as either mild or moderate to severe according to the clinical examination and electroencephalographic (EEG) data. The clinical signs of encephalopathy were graded from the first to the sixth hours of life according to the modified classification of Sarnat (i.e., mild, moderate or severe) [19]. Additionally, the initial EEG background was graded from zero to four by trained electrophysiologists according to the classification published by Murray et al. [20]. Infants were monitored with a 10-channel standard EEG device following admission to the neonatal intensive care unit, and for 24 h in non-cooled infants and during TH and rewarming in cooled infants. Thus, NE was considered mild when mild encephalopathy according to the Sarnat classification was associated with a low grade EEG background (i.e., grade 0 or 1). In other cases, NE was considered as moderate to severe.

### 2.2. Plasma Magnesium

Our unit guidelines regarding infants under TH recommend measurements of plasma total magnesium in mmol/L on admission to the unit and at approximately the 12th, 24th, 48th and 72nd hour of life. Venous blood samples were immediately shipped to the laboratory at room temperature, and magnesium measurements within plasma were performed within 4 h. No specific supplementation with magnesium was recommended. The parenteral solution included magnesium lactate at a concentration of 0.4 mg/mL. Five time periods were considered for the analysis of plasma total magnesium (i.e., from birth to six hours of life, from six to 12 h of life, from 12 to 24 h of life, from 24 to 48 h of life, and from 48 to 72 h of life).

No woman was treated with magnesium sulphate in the hours preceding the infant’s birth in our cohort.

### 2.3. Primary Outcome: Brain MRI Score

The severity of hypoxic-ischaemic brain lesions was assessed on T1- and T2-weighted brain MRI sequences performed according to the classification of Barkovich [4]. The Barkovich score includes two scores based on the grading of lesions within the basal ganglia (0 to 4) and white matter (0 to 5). MRI scoring was performed by blinded and independent observers (GF, BM). In case of disagreement between observers, the decision was made by consensus. Brain lesions on MRI were reported according to two analyses of the Barkovich score as previously proposed by Bonifacio et al. [21]. First, a dichotomous classification from the Barkovich score, i.e., low MRI score and high MRI score, was used. A low MRI score corresponded to normal MRI or mild brain lesions defined by a grey matter score less than or equal to 1 and a white matter score less than or equal to 2. A high MRI score corresponded to moderate to severe brain lesions defined by a grey matter score strictly more than 1 or a white matter score strictly more than 2 [21]. This classification was built on the results of the TOBY trial that linked high grades of brain lesions to the neurodevelopmental outcome at 18 months of age [3]. In a second set of analyses, the association between plasma magnesium and the predominant injured brain area was also explored [21]. Three patterns of injury were considered from the Barkovich score: (i) normal (both scores graded at 0), (ii) white matter (WM) injury pattern (white matter score higher than basal ganglia score) and (iii) basal ganglia/thalamus (BGT) injury score (basal ganglia/thalamus score higher than white matter score or both maximum scores) [21]. The median age at brain MRI was 5 days (*n* = 125, interquartile range: 2, range: 2–66) in our population.

### 2.4. Miscellaneous Data

Maternal age, parity, outborn birth, gestational age, sex, birth weight, and head circumference at birth were collected. Small for gestational age infants referred to infants with a birth weight strictly less than the 10th percentile according to Fenton growth charts [22]. Resuscitation actions such as intubation, chest compressions, and epinephrine administration were also reported, along with the ten-minute Apgar score. Although no standardized assessment of the subsequent neurodevelopment was available, infants were classified into three neurological outcome groups from the follow-up reports: (i) normal, (ii) alterations of neurodevelopment but independent walk is present, (iii) severe neurological impairment including cerebral palsy.

### 2.5. Statistical Analysis

In the event of normal distribution and variance equality, quantitative variables are expressed as the means (±standard deviation), and Student’s *t*-test for independent samples was used to compare groups in univariate analysis. Concerning variables with nonnormal distribution, the results are expressed as the medians (range), and the Mann–Whitney test was applied. Qualitative variables were compared using the chi-square test. The plasma magnesium kinetics were analysed using *t*-tests comparing both conditions for each time-period. The ability of plasma magnesium to predict the severity of brain injury was assessed using the receiver operating characteristic (ROC) analysis. The comparison of more than two groups of data was performed through one-way ANOVA associated with Bonferroni’s post test. A two-way ANOVA with Bonferroni’s post test was performed to explore how plasma magnesium values can be affected by perinatal factors and the MRI results. The significance threshold was strictly less than 5% (*p* < 0.05) for all analyses. Statistical analyses were performed using GraphPad Prism statistical software (version 5.01, San Diego, CA, USA) and MedCalc statistical software (version 19.1, Ostend, Belgium).

## 3. Results

### 3.1. Study Population

After the selection of eligible patients, 125 infants were ultimately included (Figure 1). Moderate to severe encephalopathy was diagnosed in 95 infants (76%) and 88 (93%) of these infants experienced TH. In contrast, only nine patients with mild encephalopathy (30%) were cooled (Figure 1).

Twelve infants (9.6%) died during the neonatal period, and 35 infants (28%) were lost to follow-up. Among the 52 surviving infants with low MRI scores and available follow-up data, 43 (82.7%) had normal neurodevelopment, seven (13.5%) exhibited mild to moderate alterations in neurodevelopment, and two (3.8%) had severe impairment. Regarding infants with high MRI scores (*n* = 26), five (19.2%) had normal neurodevelopment, 11 (42.3%) showed independent walking with mild to moderate neurodevelopmental alterations, and 10 (38.5%) had severe impairment.

### 3.2. Infants with Moderate to Severe Brain Lesions Showed a Particular Trend of Plasma Magnesium during the First 24 h of Life

The kinetics of plasma magnesium over the first 72 h of life according to the severity of MRI score are depicted in Figure 2a. Plasma magnesium was higher in infants with a high MRI score before the sixth hour of life (hour of measurement, mean (SD): 5 h (1.9)) (low MRI score (*n* = 48): 0.80 mmol/L (0.08) vs. high MRI score (*n* = 14): 0.90 mmol/L (0.16), *p* = 0.02). The infants with a high MRI score conversely showed lower plasma magnesium values than infants with a low MRI score between 12 and 24 h of life (hour of measurement, mean (SD): 21 h (5.1)) (low MRI score (*n* = 68): 0.76 mmol/L (0.1) vs. high MRI score (*n* = 37): 0.71 mmol/L (0.1), *p* = 0.01). Although plasma magnesium rose beyond the first 24 h of life in both groups, plasma magnesium remained lower in the high MRI score infants afterwards (Figure 2a). Unfortunately, the plasma magnesium value at each time interval showed a low ability to predict brain lesion severity (0–6 h period: AUC 0.69, 95% CI [0.57–0.8], 12–24 h period: AUC 0.64, 95% CI [0.54–0.73]).

Infants with a high MRI score experienced a significant drop in plasma magnesium for the first 24 h of life that was not observed in infants with a low MRI score (low MRI score (*n* = 43): −0.01 mmol/L (0.1) vs. high MRI score (*n* = 14): −0.2 mmol/L (0.15), *p* < 0.0001, Figure 2b).

### 3.3. The Injury Pattern on Brain MRI Is Associated with Early Plasma Magnesium Values

According to the Barkovich classification, 68 infants (54.4%) had normal MRI, 40 infants (32%) exhibited a WM injury pattern, and 17 infants (13.6%) showed a BGT injury pattern or a total brain injury pattern (Table 1).

Twenty-eight infants (70%) with a WM injury pattern and all infants with a BGT injury pattern had a high MRI score.

Plasma magnesium values were analysed according to the brain injury pattern during the two time periods identified above as informative for indicating the severity of brain lesions, i.e., 0–6 h and 12–24 h. Intriguingly, higher plasma magnesium during the 0–6 h time-period was significantly associated with the WM injury pattern (mean (SD), normal MRI (*n* = 38): 0.78 mmol/L (0.06), WM injury pattern (*n* = 18): 0.91 mmol/L (0.15), BGT and total brain injury pattern (*n* = 6): 0.76 mmol/L (0.06), *p* < 0.0001) (Figure 3). Plasma magnesium values during the 0–6 h time-period were a reasonable predictor of WM lesions on MRI (AUC 0.78, 95% CI [0.66–0.88], threshold value > 0.88 mmol/L, sensitivity 55.6%, specificity 93.3%). Furthermore, plasma magnesium values were similar whatever the MRI score was in infants with predominant WM injury patterns (low MRI score (*n* = 7), mean: 0.87 mmol/L (0.1) vs. high MRI score (*n* = 11), mean: 0.94 mmol/L (0.17), *p* = 0.32). These results suggest that plasma magnesium during the 0–6 h time-period was likely to be more indicative of an injury within WM than moderate to severe brain lesions.

Conversely, plasma magnesium was lower during the 12–24 h time-period in infants with brain lesions than in infants with normal MRI. The statistical threshold was only reached in infants with BGT or total brain injury patterns in comparison with infants with normal MRI (*p* = 0.04) (Figure 3).

### 3.4. Perinatal Factors and Neonatal Plasma Magnesium

The perinatal characteristics of our study population are reported in Table 2 according to the MRI-score level, i.e., low vs. high.

The context of perinatal asphyxia, i.e., acute or subacute, was not associated with the 0–6 h plasma magnesium values (acute perinatal asphyxia (*n* = 22), mean: 0.83 mmol/L (0.1) vs. subacute perinatal asphyxia (*n* = 39), mean: 0.80 mmol/L (0.09), *p* = 0.2).

The requirement of cardiac resuscitation in the delivery room, i.e., chest compressions and/or epinephrine, was associated with a higher mean value of plasma magnesium in the 0–6 h time period (no cardiac resuscitation, (*n* = 37), mean: 0.78 mmol/L (0.06) vs. cardiac resuscitation, (*n* = 24), mean: 0.86 mmol/L (0.12), *p* = 0.001). The associations of the 0–6 h plasma magnesium values with cardiac resuscitation and the MRI score were analysed through two-way ANOVA. No interaction was found between cardiac resuscitation and the MRI score. Cardiac resuscitation remained strongly associated with the increase in plasma magnesium during the 0–6 h period (*p* = 0.003), whereas the MRI score did not (*p* = 0.27). Regarding the predominant injury pattern, the same statistical test was performed. The WM injury pattern and cardiac resuscitation were independently associated with the highest plasma magnesium values (*p* = 0.0003 and *p* = 0.005, respectively).

No association was found between the severity of clinical encephalopathy according to the Sarnat classification and the plasma magnesium level during the 0–6 h time period (mild encephalopathy, (*n* = 17), mean: 0.79 mmol/L (0.06) vs. moderate to severe encephalopathy, (*n* = 43), mean: 0.82 mmol/L (0.1), *p* = 0.47).

No association was found between the initial grade of the EEG background and the plasma magnesium level during the 0–6 h time period (grade 0–1, (*n* = 16), mean: 0.79 mmol/L (0.08) vs. grade 2 or more, (*n* = 45), mean: 0.83 mmol/L (0.12), *p* = 0.27) or during the 12–24 h period (grade 0–1, (*n* = 30), mean: 0.75 mmol/L (0.1) vs. grade 2 or more, (*n* = 73), mean: 0.73 mmol/L (0.1), *p* = 0.5).

Seizures occurred much more frequently in infants with a high MRI score (*p* < 0.0001, Table 2). The median hours of life at diagnosis were 8 h 15 min [1–18] in the low MRI score group and 7 h [1–60] in the high MRI score group (*p* = 0.9). Regarding the predominant injury pattern, electric seizures occurred in 20 infants with normal MRI (29.4%), in 28 infants with WM injury patterns (70%), and in 14 infants with BGT and total brain injury patterns (82.3%). Interestingly, the occurrence of seizures was associated with a lower mean plasma magnesium value measured during the 12–24 h period (no seizure (*n* = 49), mean: 0.77 mmol/L (0.1) vs. seizures (*n* = 54), mean: 0.71 mmol/L (0.09), *p* = 0.005). This decrease in plasma magnesium in association with seizures remained significant only in infants with a predominant WM injury pattern (no seizure (*n* = 10), mean: 0.78 mmol/L (0.08) vs. seizures (*n* = 27), mean: 0.69 mmol/L (0.1), *p* = 0.02).

## 4. Discussion

To our knowledge, the precise kinetics of plasma magnesium in infants with NE over the first 72 h of life since the advent of the TH era have not been previously determined. This study highlighted that two time periods could be informative for plasma magnesium measurements, i.e., early after birth and around the 24th hour of life. Plasma magnesium values were not associated with the severity of clinical encephalopathy and EEG background and were barely associated with the severity of brain lesions. Intriguingly, plasma magnesium values during the 0–6 h time-period were likely to be linked to the presence of brain injuries that predominated within the WM. Early plasma magnesium also increased with the requirement of cardiac resuscitation in the delivery room. Thereafter, plasma magnesium decreased to reach a lower mean value during the 12–24 h period. This decrease was likely to be associated with the occurrence of seizures, and it was more evident in infants with the WM injury pattern in our cohort.

The plasma magnesium kinetics observed in our cohort did not support the data published in the early 2000s by Ilves et al. before TH implementation [16]. Magnesium was lower in the umbilical cord blood of infants with severe NE in the Ilves cohort. Thereafter, higher values of plasma magnesium were observed during the 24–48 h period in infants with severe NE than in healthy infants and infants with mild NE [15,16]. This may be due to the sample storage as samples were frozen in Ilves’ cohort, which could have altered the results [23]. A more recent study conversely showed that the lowest value of plasma magnesium was measured around 24 h of life and was more associated with a basal ganglia injury pattern on conventional MRI, as in our cohort [24].

Intriguingly, our results highlighted that plasma magnesium and the WM injury pattern were closely related. First, they had an independent and direct statistical link. Furthermore, perinatal factors that influenced plasma magnesium, i.e., cardiac resuscitation and seizures, were statistically significant in infants with a WM injury pattern in our study.

Early high plasma magnesium was observed in infants who required cardiac resuscitation in the delivery room, i.e., in an acute condition of low systemic flow. Plasma magnesium was measured in a piglet model of pure hypoxia consisting of a low oxygen supply. This experimental schedule resulted in diffuse lesions including cortical and white matter damage that could be equivalent to the more extensive WM injury pattern [25]. Plasma magnesium progressively increased during hypoxia. Plasma magnesium peaked at the end of hypoxia before decreasing for the subsequent three hours corresponding to the latent phase. Thereafter, plasma magnesium remained roughly stable during TH, whereas a slight decrease was still observed in the normothermia group [26]. Therefore, we speculate that an early high plasma magnesium could provide evidence of a recent hypoxic-ischaemic event primarily in infants with WM injury patterns. This hypothesis needs to be confirmed through further prospective data collection.

Proton (H1) magnetic resonance spectroscopy (MRS) assesses metabolism through the peaks and ratios of metabolites in specific brain areas. Two metabolites are critical in the perinatal asphyxia context: (i) an increase in lactates reflects brain energy failure, and (ii) a decrease in N-acetyl aspartate (NAA) is associated with injured neurons. These two phenomena are related to mitochondrial dysfunction and potentially to the cellular balance of magnesium. Interestingly, the lowest plasma magnesium value around the 24th hour of life was associated with the intensity of brain metabolism disturbance in infants with NE [24]. This association was observed in particular injury patterns, i.e., concerning the BGT area or multifocal and extensive white matter injuries. These data highlight that plasma magnesium measured around the 24th hour of life may reflect secondary energy failure. This hypothesis is supported by the more profound decrease in plasma magnesium observed during the 12–24 h period in infants with seizures in our cohort.

The lack of an unquestionable difference in plasma magnesium values supports the idea that NE is a complex and multifactorial condition [18]. Current issues in NE management support the emerging concept of neurocritical care. The principal aim of this approach is to potentiate the neurological outcome of these vulnerable infants through continuous bedside monitoring of brain function, close therapeutic adjustment and the limitation of environmental aggression throughout the stay in the intensive care unit [27,28]. Biological monitoring, including screening of multiorgan failure, brain dysfunction, inflammatory response and trophic factors, was included in this global management of NE [29,30]. Although our results and their interpretation need to be confirmed and sharpened through a prospective study, plasma magnesium may be included within this multimodal monitoring in the future to better identify some subtypes of NE and to individualize therapeutic management.

Blood calcium disturbance after perinatal asphyxia and during TH could also alter blood magnesium. In the piglet model of hypoxia, the calcium kinetics showed a similar but smoother trend than magnesium [26]. In parallel, calcium and magnesium values in blood and urine were previously described in cooled infants. Results showed that blood calcium variations did not likely interact with blood magnesium values [31,32].

Previous studies have shown that injuries on brain MRI performed during the neonatal period are closely associated with neurodevelopmental outcomes in infants with NE [2,3]. Several MRI scores have been developed to precisely predict the neurological prognosis of infants with NE as early as the neonatal period. The Barkovich score was the first scale validated for the stratification of brain injury severity in relation to neuromotor outcome [4]. The Barkovich score was associated with cognitive and motor outcomes at 2 years but failed to detect language difficulties at 2 years and mild brain lesions in comparison with the more detailed Weeke score, which requires diffusion-weighted imaging and proton MRS [33,34,35]. Due to the lack of proton MRS data in our cohort, the Weeke score could not be applied for the MRI analysis.

Our study has some limitations. The analysis of plasma magnesium values and our hypothesis were weakened due to our retrospective design, which led to incomplete sets of magnesium measurements and missing data mainly for infants with high MRI scores and BGT injury patterns. Therefore, our observations and analysis need to be confirmed through a prospective study. In parallel, the outcome criterion of our study was the MRI score performed during the neonatal period and not the neurodevelopmental outcome, as no homogenous follow-up was set up in our network before 2019.

## 5. Conclusions

In conclusion, plasma magnesium showed particular trends in infants with NE associated with hypoxic-ischaemic brain lesions according to the predominance of WM injury, the requirement of cardiac resuscitation and the occurrence of seizures. Two time points for plasma magnesium were informative for analysis: (i) from birth to the sixth hour of life, (ii) around the 24th hour of life. This study supports the notion that NE is a complex and multifactorial condition that is difficult to characterize precisely in the neonatal period. Prospective studies are required to confirm these results and to assess the possible role of plasma magnesium within a multimodal screening and monitoring of infants with NE to better identify the NE subtypes and adapt early therapeutic management.

## Figures and Tables

**Figure 1 children-09-01233-f001:**
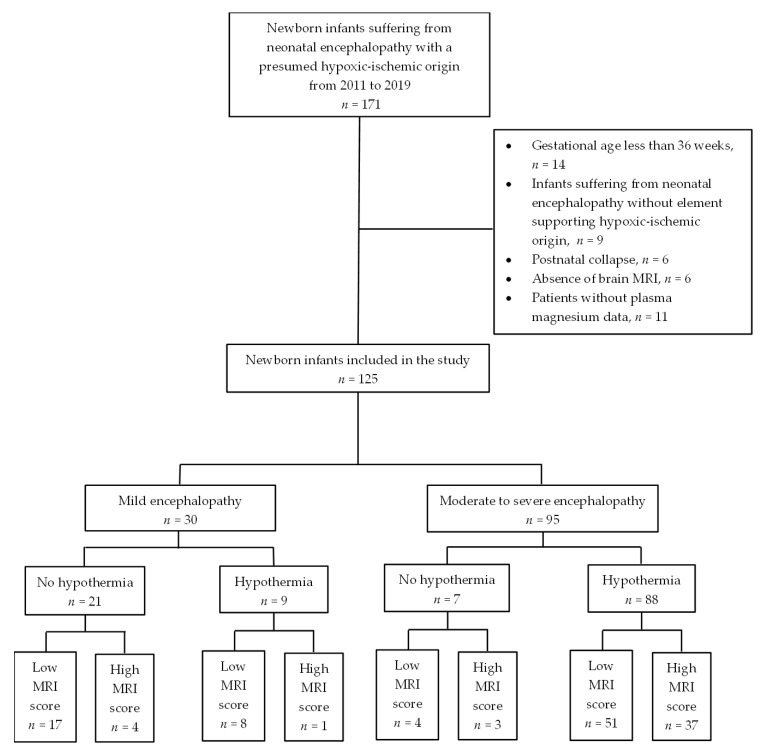
Flow-chart of the study.

**Figure 2 children-09-01233-f002:**
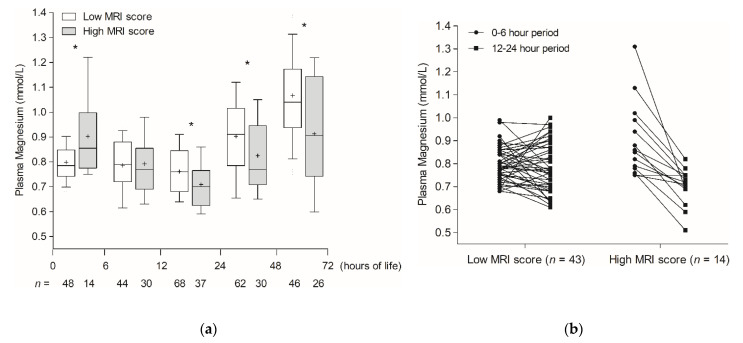
Plasma magnesium kinetics over the first 72 h of life in infants with neonatal encephalopathy according to the MRI score. (**a**) Graphic representation of plasma magnesium values (in mmol/L) according to five time periods, (i.e., from birth to the sixth hour of life, from the sixth to the 12th hour of life, from the 12th to the 24th hour of life, from the 24th to the 48th hour of life and from the 48th to the 72nd hour of life). White box-and-whiskers plots represent infants with low MRI score. Grey box-and-whiskers plots represent infants with high MRI score. Crosses in box-and-whiskers plots correspond to the means. Statistical analysis: *t*-test for independent samples, * *p* < 0.05 compared with plasma magnesium values of infants with low MRI score in the same time period. (**b**) Evolution of plasma magnesium values from the initial period, i.e., before the sixth hour of life (black circles) to the 12–24 h period (black squares) according to the MRI score.

**Figure 3 children-09-01233-f003:**
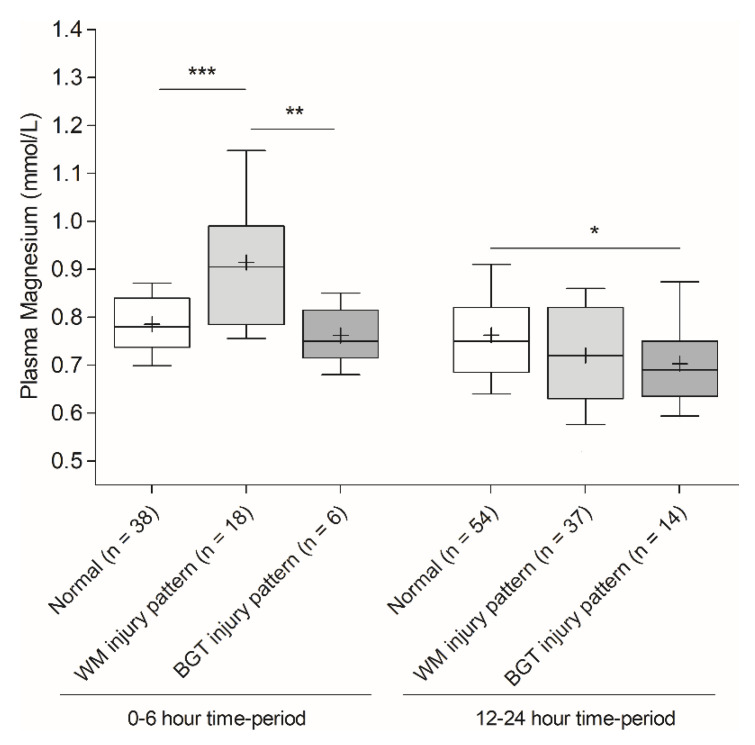
Plasma magnesium values measured before the sixth hour of life and during the 12–24 h time-period according to the brain injury pattern. White box-and-whiskers plots represent infants with normal MRI. Light grey box-and-whiskers plots represent infants with white matter injury pattern. Dark grey box-and-whiskers plots represent infants with basal ganglia-thalamus injury pattern. Crosses in box-and-whiskers plots correspond to the means. Statistical analysis: one-way ANOVA with Bonferroni’s post-test, * *p* < 0.05, ** *p* < 0.01 and *** *p* < 0.001. WM: White matter, BGT: Basal ganglia-thalamus.

**Table 1 children-09-01233-t001:** Brain lesions classified through the Barkovich MRI score according to the brain injury pattern.

	Normal (*n* = 68)	WM ^1^ Injury Pattern (*n* = 40)	BGT ^2^ Injury Pattern or Total Brain Pattern (*n* = 17)
Grey matter			
0: Normal or isolated focal cortical infarct	68	24	0
1: Abnormal signal in thalamus	0	9	0
2: Abnormal signal in thalamus and lentiform nucleus	0	5	5
3: Abnormal signal in thalamus, lentiform nucleus, and perirolandic cortex	0	2	7
4: More extensive involvement	0	0	5
White matter			
0: Normal	68	0	7
1: Single focal infarction/punctuate lesions	0	8	1
2: Abnormal signal in anterior or posterior watershed white matter	0	5	4
3: Abnormal signal in anterior or posterior watershed cortex and white matter	0	8	3
4: Abnormal signal in both anterior and posterior watershed zones	0	16	1
5: More extensive cortical involvement	0	3	1
Other brain lesions			
Brainstem injury	1	6	9
Cerebellar injury	2	3	0
Perinatal stroke	0	7	1
Corpus callosum, optical radiations	0	15	8
Extra-cerebral haemorrhage	9	2	0
Intra-ventricular haemorrhage	3	7	0

^1^ WM: White matter, ^2^ BGT: Basal ganglia-thalamus.

**Table 2 children-09-01233-t002:** Patient characteristics according to the MRI score.

	Low MRI ^1^ Score (*n* = 80)	High MRI ^1^ Score (*n* = 45)	*p*
Mother and infant characteristics			
Maternal age (years), median [range]	30 [16–43]	30 [16–44]	0.51
Parity, median [range]	2 [1–7]	2 [1–8]	0.76
Gestational age (weeks ^days^), median [range]	40 ^0/7^ [36 ^0/7^–41 ^6/7^]	39 ^2/7^ [36 ^2/7^–41 ^5/7^]	0.17
Male infants, *n* (%)	45 (56)	26 (58)	0.87
Birth weight (grams), mean (SD ^2^)	3252 (527)	3055 (524)	**0.04**
Small for gestational age, *n* (%)	13 (16.2)	11 (24.4)	0.11
Birth head circumference (cm), mean (SD ^2^)	34.7 (1.6)	34.3 (1.8)	0.29
Context of delivery			
Outborn birth, *n* (%)	54 (67.5)	31 (68.9)	0.87
Acute perinatal asphyxia, *n* (%)	34 (42.5)	12 (26.7)	0.17
Arterial pH at birth, mean (SD ^2^)	6.95 (0.15)	6.98 (0.16)	0.33
Lactate (mmol/L) at birth, mean (SD ^2^)	11.7 (4.6)	14.3 (4.8)	**0.006**
Management in the birth room			
Intubation, *n* (%)	70 (87.5)	34 (75.5)	0.09
Chest compressions, *n* (%)	27 (33.7)	20 (44.4)	0.24
Epinephrine, *n* (%)	16 (20)	14 (31)	0.16
10 min Apgar score, median [range]	6 [0–10]	5 [0–10]	0.95
Encephalopathy characteristics and management			
Sarnat classification			**0.0005**
*Mild*, *n* (%)	27 (33.7)	14 (31.1)	
*Moderate*, *n* (%)	45 (56.3)	14 (31.1)	
*Severe*, *n* (%)	8 (10)	17 (37.8)	
Grade of the first EEG background			**0.0003**
*Grade 0 or 1*, *n* (%)	36 (45)	6 (13.3)	
*Grade 2 or more*, *n* (%)	44 (55)	39 (86.7)	
Electric seizures, *n* (%)	23 (28.7)	39 (86.6)	**<0.0001**
Therapeutic hypothermia, *n* (%)	59 (73.7)	38 (84.4)	0.17
Neonatal outcome			
Neonatal death, *n* (%)	0 (0)	12 (26.6)	**<0.0001**

^1^ MRI: Magnetic resonance imaging, ^2^ SD: Standard deviation.

## Data Availability

Data are available on request to the corresponding author.

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
