# Peer review of "Early Plasma Magnesium in Near-Term and Term Infants with Neonatal Encephalopathy in the Context of Perinatal Asphyxia"

_children, 2022, doi:10.3390/children9081233_

Round 1

Reviewer 1 Report

The study aims to assess the association and predictive ability of plasma magnesium measurement and the severity of brain injuries, according to an MRI-based score, in a French cohort of neonates with neonatal encephalopathy treated or not with hypothermia.

Although the manuscript appears generally clearly written and addresses an interesting question, I have some concerns regarding the timing of plasma magnesium measurements and, consequently, the statistical analysis.

The explanation given in paragraph 2.2 (page 3) about timing of plasma magnesium evaluation is not fully clear to me. The recommendation is to measure the values “at admission and approximately at 12, 24, 48 and 72 hours from birth”. Thus, I expect that the values are taken as closest as possible to this time-points. However, in the results (paragraph 3.1, page 5, from line 182) it is reported a mean time of the first measurements of 5 with a SD of 1.9. Assuming a normal distribution this would mean that this time roughly range from 1 to 9 hours of life (approximately 95% observations should fall in the interval mean +- 2SD). Similarly, measures taken in the second period roughly range from 11 to 31 hours of life.

Given this apparent heterogeneity in the timing of plasma magnesium measures, I believe that a random-effects logistic regression model, accounting for the exact time of when measures were taken, would be more appropriate to assess the association between magnesium trend and brain injury severity.

Reviewer 2 Report

The authors retrospectively collected data from 171 term and near-term newborns treated from 2011 – 2019 in their department for neonatal encephalopathy most probably attributable to perinatal hypoxia-ischemia. 125 of these infants fulfilled the inclusion criteria for this study, most importantly that their files contained regular measurements of serum-magnesium levels between 0 and 72 hours of life and T1- and T2-MRI investigations performed in this period of time. Most of the infants were treated with hypothermia. According to clinical (Sarnat-Score) and EEG criteria 30 patients were classified as mild and 95 as moderate-severe encephalopathy. According to Barkovich criteria the MRI findings of 80 infants were scored low (good prognosis) and 45 as high (worse prognosis). In their statistical analyses the authors show that during the first 6 hours of life the magnesium values in the high MRI-score group were significantly higher than in the low score group (but only 1 or 2 exceeded the normal range for age), and that they dropped significantly stronger until the 24-48 h period (again only 1 or two dropped below the normal range). Using ROC analysis, the authors show that a drop of serum-magnesium by more than -0.11 mmol/l from the first 6 hours to the 24-48 h period statistically predicts a high MRI-score (and thus, a worse prognosis) with high sensitivity and specifity.

This is a very interesting study, to my knowledge only very few investigations in serum-magnesium in asphyxiated newborn infants have been published so far. However, as also the authors are aware of, the study is compromised by its retrospective design which could lead to systematic biases. There are some points to be cleared before a decision on a publication is possible:

1.       Why have only the magnesium values been included in the study? There is a role of magnesium in the pathogenesis of hypoxic-ischemic encephalopathy, but even more is known on the role of calcium and glucose which can also drop in severe asphyxia. The role of excitotoxic influx of calcium into the neuronal cytosol and mitochondria is probably as least as important as the influx of magnesium.

2.       Magnesium has played a great role in obstetrics in treatment of eclampsia and premature labor, and it has been discussed (and applied by some) as protection against hypoxic-ischemic brain damage. Thus, information is needed on a possible treatment of the mothers with magnesium sulfate which could declare the high initial values in some children.

3.       They write that on their NICU “No specific supplementation with magnesium was recommended.” But was it strictly prohibited? The files should show if some children were treated with magnesium against the recommendation.

4.       The analyses are restricted to the magnesium kinetics and the MRI findings. However, correlation of serum magnesium with the clinical score and the EEG score or multivariable analyses could also be of value.

5.       The patient numbers in figure 2 do not correspond to the 80 patients with low and 45 with high MRI-score. Namely, in the high MRI score group in figure 2b) there are only 14 cases! This is apparently an analysis on a highly selected group of cases which does not allow any interpretation!

6.       Before any recommendation to use the magnesium kinetics in the first days as a biomarker, a repetition of the study in a prospective cohort would be necessary.

7.       The authors should try to gather at least some semi-quantitative information on the clinical outcome of the children (normal – borderline/slight) severe handicapped).

There are some weaknesses in the English language, some of them are:

1.       Abstract: “The kinetics of plasma magnesium was performed for the first 72 hours of life according to the Barkovich MRI score” should better read “The kinetics of plasma (or serum?) magnesium concentration was investigated for the first 72 hours of life and correlated to the Barkovich MRI score”.

2.       Abstract: “Plasma magnesium dosing at both these time-points”: probably they mean plasma (or serum?)  magnesium “concentrations”.  “Dosing” would be appropriate for a drug dosage.

3.       Line 72: “kinetic” is an adjective, the noun is “kinetics”.

4.       Line 75/76 and later in text (results, discussion): “to predict the severity and the pattern of brain injury”. This sentence suggests that this is a prognostic study that can lead to conclusion on future findings and prognosis. However, in my view this is only a correlative/association study of two biomarkers: the magnesium kinetics and the MRI-score, apparently both already exist at the same time in the pathophysiological process. It is only a statistical “prediction” in the ROC analysis as shown in line 158.

5.       Line 181: better depicted than reported

6.       Line 281: lactates: why the plural?

Author Response

Reviewer 2

Comments and Suggestions for Authors

The authors retrospectively collected data from 171 term and near-term newborns treated from 2011 – 2019 in their department for neonatal encephalopathy most probably attributable to perinatal hypoxia-ischemia. 125 of these infants fulfilled the inclusion criteria for this study, most importantly that their files contained regular measurements of serum-magnesium levels between 0 and 72 hours of life and T1- and T2-MRI investigations performed in this period of time. Most of the infants were treated with hypothermia. According to clinical (Sarnat-Score) and EEG criteria 30 patients were classified as mild and 95 as moderate-severe encephalopathy. According to Barkovich criteria the MRI findings of 80 infants were scored low (good prognosis) and 45 as high (worse prognosis). In their statistical analyses the authors show that during the first 6 hours of life the magnesium values in the high MRI-score group were significantly higher than in the low score group (but only 1 or 2 exceeded the normal range for age), and that they dropped significantly stronger until the 24-48 h period (again only 1 or two dropped below the normal range). Using ROC analysis, the authors show that a drop of serum-magnesium by more than -0.11 mmol/l from the first 6 hours to the 24-48 h period statistically predicts a high MRI-score (and thus, a worse prognosis) with high sensitivity and specifity.

This is a very interesting study, to my knowledge only very few investigations in serum-magnesium in asphyxiated newborn infants have been published so far. However, as also the authors are aware of, the study is compromised by its retrospective design which could lead to systematic biases. There are some points to be cleared before a decision on a publication is possible:

  1. Why have only the magnesium values been included in the study? There is a role of magnesium in the pathogenesis of hypoxic-ischemic encephalopathy, but even more is known on the role of calcium and glucose which can also drop in severe asphyxia. The role of excitotoxic influx of calcium into the neuronal cytosol and mitochondria is probably as least as important as the influx of magnesium.

Authors’ response to reviewer:

I agree with that comment. Our study aim was to focus on plasma magnesium. Unfortunately, the retrospective design did not allow to analyze the complex relation between plasma calcium and magnesium. Experimental and clinical data reported the kinetics of blood calcium and blood magnesium after a hypoxic-ischaemic insult. As alteration of blood calcium and magnesium were observed, the influence of blood calcium on the blood magnesium level was not evident. A comment was added in the discussion section about this point: page 10 line 369-373.

  1. Magnesium has played a great role in obstetrics in treatment of eclampsia and premature labor, and it has been discussed (and applied by some) as protection against hypoxic-ischemic brain damage. Thus, information is needed on a possible treatment of the mothers with magnesium sulfate which could declare the high initial values in some children.

Authors’ response to reviewer:

 I agree with that comment. However, no women were treated with magnesium sulphate just before birth in our cohort. A sentence has been added in the method section (paragraph 2.2, page 3, line 127-128):No woman was treated with magnesium sulphate in the hours preceding the infant birth in our cohort’.

  1. They write that on their NICU “No specific supplementation with magnesium was recommended.” But was it strictly prohibited? The files should show if some children were treated with magnesium against the recommendation.

Authors’ response to reviewer:

 I agree with that comment. The retrospective design did not allow us the complete control of this parameter. However, the initial intravenous solution contained magnesium lactate at a concentration of 0.4 mg/ml. On the second day, physicians could prescribe an individualized parenteral solution if needed. Although, there is no recommendation about plasma magnesium level and its potential correction, data files report that physicians could prescribe a supplementation by magnesium sulphate (usually 5 mg/kg) in the daily parenteral solution on the basis of the H12-H24 measure of plasma magnesium. In parallel, the early kinetics observed is comparable to the kinetics observed in the piglet model without any supplementation.

  1. The analyses are restricted to the magnesium kinetics and the MRI findings. However, correlation of serum magnesium with the clinical score and the EEG score or multivariable analyses could also be of value.

Authors’ response to reviewer:

A third subsection has been added in the result section to explore the link between plasma magnesium and perinatal factors as the type of asphyxia, the cardiac resuscitation, the initial EEG background, the initial grade of encephalopathy and seizures.(pages 7-9, lines 264-305).

  1. The patient numbers in figure 2 do not correspond to the 80 patients with low and 45 with high MRI-score. Namely, in the high MRI score group in figure 2b) there are only 14 cases! This is apparently an analysis on a highly selected group of cases which does not allow any interpretation!

Authors’ response to reviewer:

Due to retrospective design of the study, patients had incomplete set of data. This explains the few number of data for this analysis. Therefore, the ROC analysis was retrieved of the manuscript.

  1. Before any recommendation to use the magnesium kinetics in the first days as a biomarker, a repetition of the study in a prospective cohort would be necessary.

Authors’ response to reviewer:

Authors agree with that comment and recognized that they overstated the results of this study. The conclusion section, some parts of the discussion section and abstract have been rewritten accordingly.

  1. The authors should try to gather at least some semi-quantitative information on the clinical outcome of the children (normal – borderline/slight) severe handicapped).

Authors’ response to reviewer:

These information have been added in the manuscript under a descriptive form.

In the method section: page 4, lines 156-160

In the result section: page 5, lines 185-191

There are some weaknesses in the English language, some of them are:

  1. Abstract: “The kinetics of plasma magnesium was performed for the first 72 hours of life according to the Barkovich MRI score” should better read “The kinetics of plasma (or serum?) magnesium concentration was investigated for the first 72 hours of life and correlated to the Barkovich MRI score”.

Authors’ response to reviewer:

Authors agree with this comment. This sentence has been modified.

  1. Abstract: “Plasma magnesium dosing at both these time-points”: probably they mean plasma (or serum?) magnesium “concentrations”.  “Dosing” would be appropriate for a drug dosage.

Authors’ response to reviewer:

Authors agree with this comment. This sentence has been modified.

  1. Line 72: “kinetic” is an adjective, the noun is “kinetics”.

Authors’ response to reviewer:

Authors agree with this comment. This word has been modified all along the manuscript.

  1. Line 75/76 and later in text (results, discussion): “to predict the severity and the pattern of brain injury”. This sentence suggests that this is a prognostic study that can lead to conclusion on future findings and prognosis. However, in my view this is only a correlative/association study of two biomarkers: the magnesium kinetics and the MRI-score, apparently both already exist at the same time in the pathophysiological process. It is only a statistical “prediction” in the ROC analysis as shown in line 158.

Authors’ response to reviewer:

Authors agree with that comment. Authors agree that they overstated their results. The meaning of our results has been readjusted throughout the manuscript.

  1. Line 181: better depicted than reported

Authors’ response to reviewer:

This word has been modified.

  1. Line 281: lactates: why the plural?

Authors’ response to reviewer:

This word has been modified.

Round 2

Reviewer 1 Report

The authors provided results of the association between plasma magnesium levels and MRI-score group adjusting for the exact timing of magnesium measurement, using simple logistic regression (although a random-effects model, accounting for the within-subjects correlations, would have been more appropriate).

Reviewer 2 Report

The authors have accepted most of this reviewers comments and changed their manuscript accordingly. Most importantly, they have dampened the interpretation of their retrospective data and recommended further prospective investigations before valid conclusions for prognostication and treatment could be drawn.

Myself not being a native English speaker, I nevertheless found some incorrect wording, as described below. I recommend anew editing of the text by a native speaker.

Line 186: "beyond" would mean outside of this group of patients. I believe what they mean is "among"

Line 313: "likely to be closely linked" sounds like a mixture of a cautious expression and a strong believe. I would propose either "likely to be linked" if they are not quite sure, or "closely linked" when they are sure.

Line 331: "were more meaningful" sounds to me like "makes more sense". I believe they want to say "plays a greater role" or "was more frequent"

Line 373: "did not likely to interact": the "to" is not correct in this place
